# Use of AbobotulinumtoxinA for Cosmetic Treatments in the Neck, and Middle and Lower Areas of the Face: A Systematic Review

**DOI:** 10.3390/toxins13020169

**Published:** 2021-02-22

**Authors:** Hassan Galadari, Ibrahim Galadari, Riekie Smit, Inna Prygova, Alessio Redaelli

**Affiliations:** 1College of Medicine and Health Sciences, United Arab Emirates University, Al Ain P.O. Box 17666, UAE; galadari@email.com; 2Riekie Smit Practice, Pretoria 0182, South Africa; riekiesmit@icloud.com; 3Ipsen Pharmaceutical, 92100 Boulogne-Billancourt, France; inna.prygova@galderma.com; 4Visconti di Modrone Medical Center, 20122 Milan, Italy; mail@docredaelli.com

**Keywords:** abobotulinumtoxinA, botulinum toxin, lower face, marionette lines, masseter, neck, perioral area, platysma, middle face

## Abstract

AbobotulinumtoxinA (aboBoNT-A) has been used for various cosmetic purposes, including minimization of moderate to severe lines, or other cosmetic indications, in the face and neck. We carried out a systematic review to identify all relevant evidence on the treatment approaches and outcomes of aboBoNT-A as a cosmetic treatment of the middle and lower areas of the face, and the neck. Embase, MEDLINE, Cochrane Library, congress proceedings and review bibliographies were searched for relevant studies. Identified articles were screened against pre-specified eligibility criteria. Of 560 unique articles identified, 10 were included for data extraction (three observational studies, 1 randomized controlled trial [with two articles] and five non-randomized trials). The articles provided data on gummy/asymmetric smile (2), marionette lines (5), masseter muscle volume (2), nasal wrinkles (2), perioral wrinkles (3) and the platysma muscle (4). All articles reporting on efficacy of aboBoNT-A demonstrated positive results, including reduction of wrinkles (5), reduction of masseter muscle (2) and degree of gummy smile (1) compared with before treatment. No serious adverse events were reported and patient satisfaction was high. In conclusion, positive findings support further research of aboBoNT-A for the middle and lower areas of the face, and in the neck, which are largely unapproved indications.

## 1. Introduction

Botulinum neurotoxin type-A injections are currently the most popular type of cosmetic procedure worldwide [1,2,3,4]. The neurotoxin reduces muscle contractions by blocking the release of the neurotransmitter acetylcholine and can be used to minimize wrinkles and other conditions, caused by repeated movements and muscle contractions [5,6]. There are several preparations of botulinum neurotoxin type-A available for aesthetic use [6,7,8]. Three of the well-known preparations, which have different manufacturing processes and properties, are available worldwide: abobotulinumtoxinA (aboBoNT-A; Dysport^®^, Ipsen Biopharm Ltd., Wrexham, UK; Azzalure^®^, Galderma Ltd., Lausanne, Switzerland), onabotulinumtoxinA (onaBoNT-A; Botox^®^, Allergan, Irvine, CA, USA), and incobotulinumtoxinA (incoBoNT-A; Xeomin^®^, Merz Pharmaceuticals GmbH, Frankfurt, Germany) [9,10,11,12]. 

For aesthetic treatment, aboBoNT-A (Dysport 300 unit vial [Speywood units]) is approved in the USA by the Food and Drug Administration for the temporary improvement in the appearance of moderate to severe glabellar lines associated with procerus and corrugator muscle activity in adult patients less than 65 years of age [10]. In the EU, it is indicated (Azzalure 125 unit vial) for the temporary improvement in the appearance of moderate to severe glabellar lines seen at frown and/or lateral canthal lines seen at maximum smile, in adult patients less than 65 years, when the severity of these lines has an important effect on the psychological well-being of the patient [13]. It is approved in many other countries worldwide for glabellar lines and other aesthetic indications, although it is common for physicians to use aboBoNT-A to treat wrinkles and lines in other areas of the face and neck [14,15]. Several consensus recommendations for the use of aboBoNT-A for treatment of wrinkles or other cosmetic indications on the middle and lower face, and neck and chest have been developed [4,14,16,17]. These recommendations provide information on the number and location of the injection points, the dose of individual injection points, the combined dose of multiple injection points, the injection technique and safety concerns for each area [4,14,16,17]. 

Owing to its approved indications, most research on the cosmetic use of aboBoNT-A is in relation to glabellar lines and the upper area of the face [18,19,20]. As such, there is a paucity of research on the use of aboBoNT-A in the middle and lower regions of the face and in the neck. The aim of this systematic literature review (SLR) was to assess the evidence regarding treatment approaches, efficacy, safety and patient-reported outcomes relating to aboBoNT-A for cosmetic treatment for several indications for which aboBoNT-A is largely unapproved, namely, treatment of the neck and of middle and lower areas of the face.

## 2. Results

### 2.1. Systematic Review of the Literature

A total of 625 papers were identified in the electronic database searches; 65 duplicate papers were excluded before citation screening and 510 papers were excluded during citation screening (Figure 1). There were no relevant studies identified in the supplementary searches, and 40 of the remaining 50 papers were excluded during full-text review, leaving 10 studies (three observational studies [OSs] [21,22,23], five non-randomized controlled trials [24,25,26,27,28] and two studies reporting data from one randomized controlled trial [RCT] [29,30]) included in the SLR. In the included studies, sample size ranged from 10 to 383 patients, mean age was 32.8 to 55.9 years and percentage of female participants was 92.7% to 100% across six studies that reported gender. Three studies were conducted in Brazil [22,29,30], two in Lebanon [24,26], two in the USA [21,27] and one each in South Korea [25], Taiwan [28] and Thailand [23]. Six of the studies looked at aboBoNT-A injections in various parts of the middle and lower regions of the face for the treatment of various conditions, including lower eyelid wrinkles, nasal wrinkles, platysmal bands at rest and at maximal contraction and marionette lines [23,24,26,27,29,30]. Of the remaining four studies, two looked at patients that received injections in the masseter muscles [25,28], one looked only at treatments for excessive gingival display (gummy smile) [22] and one looked at injections in the platysma muscle only [21].

### 2.2. Treatment Approaches

Treatment approaches are summarized, by study, in Table 1, and by indication, in Table 2 and Figure 2. All except one study [27] included approaches to toxin dilution. In eight studies, aboBoNT-A was diluted in saline to produce concentrations ranging from 70 to 250 U/mL (most often 2 or 2.5 mL normal saline to dilute 500 U of aboBoNT-A) for different indications (Figure 2) [21,22,23,24,25,26,29,30]. In one study, aboBoNT-A was diluted in sterile distilled water to produce a concentration of 200 U/mL for the reduction of masseter muscle volume [28]. Table 1 provides details of the specific areas injected in each study. Participants were injected in one [22,27,28] or two [24,25,29,30] treatment sessions; three studies did not report injection frequency [21,23,26]. Only one study reported the use of topical anaesthesia, in which patients were treated with a cream containing lidocaine and prilocaine before injection [22]. The number of injection points varied with area of treatment, ranging from two for treatment of anterior gummy smile [22] to 12 for injections into the platysma muscle [21]. In one study looking at injections for rejuvenation of the neck using a multiple-injection technique, participants received approximately 150 micro injections (mean total dose of 154 U aboBoNT-A) across the anterior region of the neck [26].

### 2.3. Efficacy Outcomes

Of the included studies, eight reported efficacy outcomes associated with aboBoNT-A treatment to the neck or lower and middle areas of the face (Table 3) [22,23,24,25,26,27,28,30]. In six studies that reported timing of clinical effects, improvement was observed at different follow-up times for different treatment areas [23,24,25,27,28,30]. In one study looking at aboBoNT-A treatment in mid-face lifting (involving injection in the platysma muscle and orbicularis oculi), clinical effect was instantaneous in some cases, but in most cases, changes occurred at 5–10 days [23]; whereas in another study of the effect of aboBoNT-A treatment on lower eyelid wrinkles, nasal wrinkles, malar wrinkles, perioral wrinkles, marionette lines, gummy/asymmetric smile and cellulitic chin, improvement was observed at the first follow-up 4 weeks following treatment [30]. Chang et al. also reported an effect of aboBoNT-A treatment on the magnitude of strain in the cheek, marionette lines, nasolabial folds, oral commissures, upper lip and perioral region as a whole, at the first follow-up 2 weeks following treatment [27]. For reduction of masseter muscle volume, clinical effect of aboBoNT-A treatment was observed from 2 to 4 weeks [25,28], and for jowls, platysmal bands, marionette lines, neck volume and oral commissures, improvement was observed at the 15- and 30-day follow-up visits [24].

A duration of action for aboBoNT-A treatment of 3–5 months was reported for gummy smile [22], 10–14 weeks for mid-face lifting involving injection of the platysma and lateral part of the orbicularis oculi [23] and improvement was observed up to 20 weeks following aboBoNT-A treatment of wrinkles in the middle and lower face [30] and up to 90 days following aboBoNT-A treatment of marionette lines [27]. One study of aboBoNT-A treatment for reduction of masseter muscle volume also reported maximum effect at 10–12 weeks following treatment [25].

All eight studies reported improvements following aboBoNT-A treatment in the neck, and middle or lower face [22,23,24,25,26,27,28,30]. Two studies that analysed patients from the same study sample (n = 30 [100% women]) demonstrated significant improvements in scores on a photonumeric scale [31,32] when measuring wrinkles/lines in the platysmal bands at 15 days following treatment compared with before treatment [24,26]. Two studies, a non-randomized controlled trial (n = 32 [100% women]) [27] and an RCT (n = 85 [82 women; 3 men]) [30], showed reductions in the magnitude of strain in the marionette lines, perioral area and chin, at follow-up times of 2–4 weeks. Improvements were also demonstrated, in single studies, for jowls [26], lip [27], and eyelid and nasal wrinkles [30]. Two studies investigated the change in masseter muscle volume at 3 months after injection with aboBoNT-A [25,28]. Kim et al. (n = 383 [355 women; 28 men]) [25] and Yu et al. (n = 10 [100% women]) [28] both demonstrated reductions in masseter muscle volume by approximately 30%; these studies used an ultrasonogram and computed tomography (CT) scan, respectively, to measure muscle volume. Finally, one study investigated the degree of gum display in 16 participants with gummy smile [22]. Findings showed that there were improvements in anterior, posterior, mixed and asymmetric gummy smile of 61–96%, at follow-up 20–30 days following treatment.

### 2.4. Safety Outcomes

Nine studies reported on the safety outcomes associated with the injection of aboBoNT-A in the neck, and upper, middle and lower areas of the face (Table 3) [21,22,23,24,25,26,27,28,30]. The adverse events (AEs) or temporary side effects associated with aboBoNT-A treatment were mild or moderate and occurred infrequently, as follows: injection-point ecchymosis [24,26,33]; lip/face asymmetry [22,23,28,30]; difference in crunching power [25,28]; mild dysphagia following injections in the neck [24,26]; excessive perioral weakness [30]; minor neck muscle weakness [24]; unnatural smiling appearance [25] and disappearance of facial dimple [25]. One of the studies observed AEs under a novel technique, micro injections [26], and the Nefertiti technique [34]. More events of ecchymosis were observed with the Nefertiti technique than with the micro injections (n = 6 vs. n = 3) and mild dysphagia was observed with the Nefertiti technique but not with micro injections [26]. Two studies reported that there were no AEs associated with aboBoNT-A treatment [21,27]. Three studies assessed pain after injection using patient-rated scales [24,26,28]. Mean visual analog scale (VAS) scores for pain from injections ranged from 0.6 to 4.6 on scale of 0 to 10 [24,26], and a mean score of 3 was reported in one study using VAS scores of 1 to 5 [28].

### 2.5. Patient Satisfaction and Quality of Life

Six studies reported patients satisfaction outcomes [22,24,25,26,27,29] and one study assessed health-related quality of life (Appendix A: Summary of patient satisfaction and quality of life outcomes) [29]. Most studies used simple surveys, asking the patient whether they were satisfied with their results. Only one study [27] used a formal assessment that has been evaluated elsewhere: the FACE-Q survey [33,35,36]. Five studies showed that 84–100% of participants were satisfied with the results of their treatment (Figure 3) [22,24,25,26,29]. In one study, the participants were 22.2% (*p* = 0.014) more satisfied with their overall facial appearance at day 14 compared with baseline [27]. In one study that used two techniques, 72% of patients preferred micro injections, 20% preferred the Nefertiti technique and 8% had no preference [26]. In the study that assessed quality of life, which used the World Health Organization Quality of Life—Brief Version questionnaire (WHOQOL-BREF), there was a significant improvement in the physical quality of life domain between baseline and 4 weeks. When comparing total dose groups, a medium dose group (166–205 U) had significantly better physical, psychological and social relationships quality of life scores than the low (120–165 U) and high (206–250 U) dose groups [29].

## 3. Discussion

To our knowledge, this is the first published SLR describing the evidence on the treatment approaches, efficacy and safety outcomes associated with the use of aboBoNT-A specifically for cosmetic treatment of the neck, and middle and lower areas of the face.

This SLR identified several studies reporting a range of treatment approaches and provides insight into the dose, number of injection points, injection frequency and toxin dilution for aboBoNT-A treatment of the lower and middle areas of the face and the neck. A broad range of treatment approaches were reported, although there were some commonalities amongst some studies, for example in the aboBoNT-A dilution used (2 or 2.5 mL of normal saline to dilute 500U of aboBoNT-A).

A broad range of efficacy outcomes were measured in the different studies; measures to examine improvement following treatment included validated photonumeric scales, digital image correlation and clinician assessment of photographs. To enable the comparison of data across studies, further research using comparable outcome measures would be beneficial.

All studies identified in this SLR reported that aboBoNT-A treatment is effective for cosmetic use, regardless of the indication. Studies that asked if patients were satisfied with the results of aboBoNT-A treatment showed a high proportion of positive responses. It is not clear from the evidence if satisfaction is driven by the results themselves, onset or duration of results, or all of the above. In this review, five of ten studies reported duration of action for aboBoNT-A treatment and found that duration of action was 2–5 months across various indications [22,23,25,27,30]. This is similar to findings from an international expert consensus on both facial rejuvenation and primary hyperhidrosis, which stated that duration of aboBoNT-A effect was up to 4–6 months for repeated treatment, and could be more than 6 months in some cases [16]. In addition, a systematic review of aboBoNT-A treatment of the upper face found that a typical duration of action was 4 months, across 18 studies [37]. Data on pain experienced during injection were limited, with only three studies reporting this outcome. In addition, only one study examined quality of life outcomes [29].

No serious AEs associated with aboBoNT-A use were reported in any of the studies. One potential serious AE associated with cosmetic injection of botulinum toxins is dysphagia, which can occur as a result of paralysis of muscles near the injection area [38]. Only one case of dysphagia was reported among the ten studies in the current review and its severity was mild. Possibly this is owing to the dose levels among these studies, as risk of dysphagia may depend on dose [39].

A strength of this SLR is that no country and date limits were included in the protocol; therefore, the results provide evidence on the global use and treatment patterns of aboBoNT-A for the neck and lower and middle areas of the face.

However, overall, the number of studies identified in the SLR was low, with many studies identified in the electronic searches looking only at upper regions of the face and therefore being excluded during citation screening. The studies were limited to those published in English and further studies in other languages were not included, which may contribute to the amount of published evidence identified in this SLR being low. Furthermore, the low number of studies identified and the broad range of approaches reported, prevent any statistical analysis of the data combined from different studies.

The aim of this systematic review was not to generate data that would be used to seek regulatory approval for the use of aboBoNT-A as a cosmetic treatment of the neck, and middle and lower areas of the face and none of the studies included in the SLR are funded by manufacturers of aboBoNT-A. The SLR was conducted by the investigators with better scientific understanding as the main aim.

## 4. Conclusions

The findings of this SLR demonstrate that aboBoNT-A is used for a range of cosmetic treatments in the lower and middle areas of the face, and in the neck, with positive outcomes for patients and a low complication rate. However, the number of studies identified was low and a broad range of approaches was identified. Further research is needed to establish uniform protocols to allow consistent treatment approaches and optimize treatment outcomes for patients.

## 5. Materials and Methods

### 5.1. Search Strategy

The SLR was performed to identify studies relevant to at least one of the following four key areas: (i) treatment approaches, (ii) efficacy outcomes, (iii) safety outcomes and (iv) patient-reported outcomes, including patient satisfaction and quality of life. The literature searches were conducted in August 2019 in the following electronic databases: (1) MEDLINE In-Process & Other Non-Indexed Citations and OVID MEDLINE, 1946–present, (2) Embase, 1974–present and (3) Cochrane Library, comprising American College of Physicians Journal Club, Cochrane Database of Systematic Reviews, Cochrane Central Register of Clinical Trials, Evidence-Based Medicine Reviews Database of Abstracts of Reviews of Effects, National Health Service Economic Evaluation Database, Health Technology Assessment Database and all entries in Evidence-Based Medicine Reviews (Appendix A: Searches carried out in: Ovid MEDLINE, all segments, 1946 to present; Embase 1974 to 18 July 2019; and Cochrane (CDSR, DARE, CENTRAL, NHS EED, The HTA database, and ACP Journal Club) 23 August 2019). The SLR is fully compliant with the 2009 Preferred Reporting Items for Systematic reviews and Meta-Analyses (PRISMA) guidelines [40]. There is no protocol available for this SLR. The bibliographies of pertinent narrative reviews identified in the SLR and proceedings from The American Academy of Dermatology and European Society for Dermatologic Research annual congresses (2017‒2019) were searched for relevant studies.

### 5.2. Eligibility Criteria

The title and abstract of identified publications were screened manually against pre-defined eligibility criteria for each objective (Appendix A: Eligibility criteria used in the systematic literature review). The searches were limited to human studies published in English. OSs, non-randomized controlled trials and RCTs were included. Narrative reviews were included for the first screening so that the bibliographies could be searched for any relevant OSs, non-randomized controlled trials or RCTs cited; narrative reviews were excluded at full-text review. Full-text versions of all publications meeting the eligibility criteria at first screening were reviewed. Data from eligible studies were extracted manually into pre-defined summary tables.

### 5.3. Data Extraction

Information extracted from relevant studies included study design, context (aims/objectives, outcomes/endpoint, disease type, sample size, comparator group, follow-up period), participants (demographics, disease duration), treatment approaches (area, dose and dilution, number of injection points and injection frequency), efficacy outcomes (wrinkle/line assessment, gum display, global aesthetic improvement scale, onset/duration of action), safety outcomes (AEs), patient satisfaction and quality of life.

## Figures and Tables

**Figure 1 toxins-13-00169-f001:**
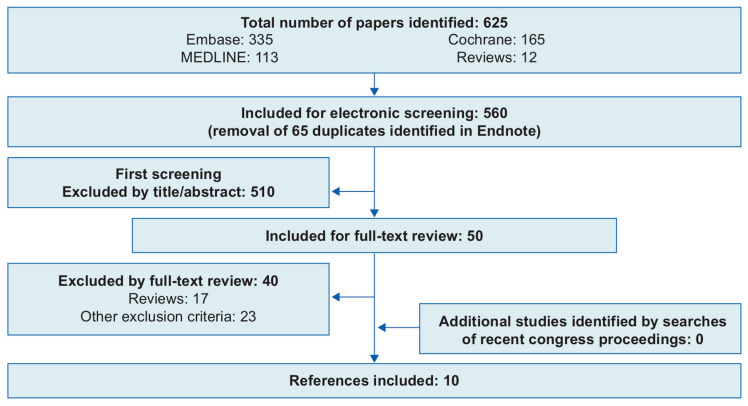
Preferred Reporting Items for Systematic Reviews and Meta-Analyses flow diagram. References were excluded at full-text review stage owing to no relevant data as follows: (1) duplicates, (2) editorial/commentary, (3) population not of interest, (4) intervention not of interest (not aboBoNT-A), (5) outcomes not of interest, (6) study design (non-randomized controlled trial/modeling study/case series or studies), (7) study not in English, (8) in vitro/animal study and (9) website source. Narrative reviews were included for first screening so that any relevant cited studies could be identified and were excluded at full-text review.

**Figure 2 toxins-13-00169-f002:**
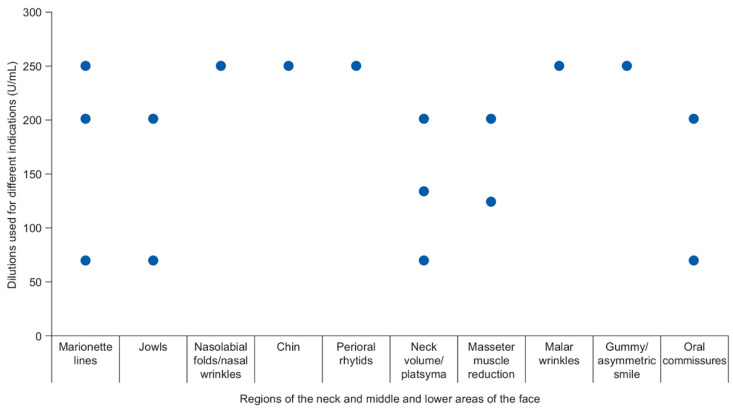
AbobotulinumtoxinA dilutions used for different indications.

**Figure 3 toxins-13-00169-f003:**
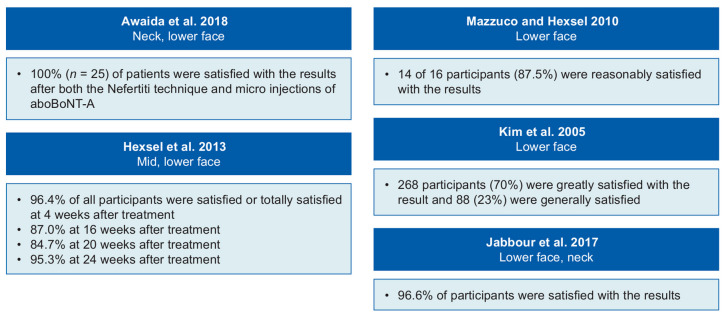
Patient satisfaction with abobotulinumtoxinA (aboBoNT-A) treatment. References: Awaida et al. [26]; Hexsel et al. [29]; Mazzuco and Hexsel [22]; Kim et al. [25]; Jabbour et al. [24].

**Table 1 toxins-13-00169-t001:** AbobotulinumtoxinA (aboBoNT-A) treatment approaches, by study.

Study	Study Design	Country	Age, Years	Participants, N (n [%] Women, n [%] Men)	Area(s) of Injection. Further Details	Indication Assessed	Total Dose and Number of Injection Points	Injection Frequency	AbobotulinumtoxinA Dilution
Awaida et al., 2018 [26]	Non-randomized controlled trial	Lebanon	Mean (SD): 55.9 (5.8)	25 (25 [100%] women)	Neck, lower face. Injections were administered over the entire anterior neck	Oral commissures, marionette lines, jowls, neck volume, platysmal bands at rest and at maximal contraction	Mean (SD) dose of 154 (28.6) U in 150 points of injection	NR	500 U vial reconstituted in normal saline to a concentration of 70 U/mL
Chang et al., 2018 [27]	Non-randomized controlled trial	USA	Mean: 51.7 Range: 28.8–72.4	32 (32 [100%] women)	Lower face. Left upper, right upper, left lower, and right lower cutaneous lip	Perioral rhytids, marionette lines, chin, nasolabial fold, oral commissures, cheeks	4–5 U per point in four points (left upper, right upper, left lower and right lower cutaneous lip; 18 U in total)	One session	NR
Hevia 2010 [21]	Observational study	USA	Mean: 50 ^1^ Range: 21.0–78.0 ^1^	43 ^2^ (NR)	Neck. Platysma	Platysma	4–12 injections (50–160 U in total)	NR	300 U was reconstituted with 2.25 mL of 0.9% saline, resulting in concentration of 133 U/mL
Hexsel et al., 2013 [30]	RCT	Brazil	Mean (SD): 48.3 (7.2) Range: 30.0–60.0	85 (82 [96.5%] women; 3 [3.5%] men)	Mid, lower face. In each third of the face, at least two of the assessed indications (see next column) were injected	Lower eyelid wrinkles, nasal wrinkles, malar wrinkles, perioral wrinkles, marionette lines, gummy/asymmetric smile, cellulitic chin	Comparison of 120‒165 U, 166‒205 U and 206‒250 U Number of injection points NR	≥2 sessions	500 U reconstituted in 2 mL of 0.9% sterile saline, resulting in 250 U/mL
Hexsel et al., 2013 [29]	RCT	Brazil		As above (Hexsel et al., 2013 [30])					
Jabbour et al., 2017 [24]	Non-randomized controlled trial	Lebanon	Mean (SD): 54.8 (5.3)	30 (30 [100%] women)	Lower face, neck. Injections administered 1–2 cm apart on a horizontal line under the mandibular border, followed by platysmal band injections 2 cm apart, vertically	Jowls, platysmal bands at rest and at maximal contraction, marionette lines, neck volume, oral commissures	125 U maximum for global neck treatment per injection session (5 U per point in 2–4 points on each platysmal bands and for mandibular border) Mean (SD) dose of 114.3 (13.7) U	Two sessions	500 U reconstituted in 2.5 mL of sterile saline
Kim et al., 2005 [25]	Non-randomized controlled trial	South Korea	Age ranges: 13‒19 years (n = 10) 20‒29 years (n = 293) 30‒39 years (n = 70) 40‒49 years (n = 9)	383 (355 [92.7%] women; 28 [7.3%] men)	Lower face. Within 1.5 cm of the mandible angle border	Masseter muscle	100–140 U on each side	1–2 injections	500 U reconstituted in 4 mL sterile saline to a final concentration of 125 U/mL
Mazzuco and Hexsel 2010 [22]	Observational study	Brazil	NR	16 (NR)	Lower face. Each side of the nasolabial fold and/or the malar region, depending on type of indication	Gummy smile (anterior/posterior/mixed)	5–15 U and 2–6 injection points depending on gummy smile type (see Mazzuco and Hexsel [22] for full details)	One session	500 U diluted in 2 mL of 0.9% sodium chloride solution
Petchngaovilai 2009 [23]	Observational study	Thailand	Range: 27.0–72.0	261 (NR)	Mid-lower face. Mid-face lifting involving injection of the platysma and lateral part of the orbicularis oculi	Mid-face, including the platysma	50–70 U per side Number of injection points NR	NR	500 U diluted in 7 mL of normal saline
Yu et al., 2007 [28]	Non-randomized controlled trial	Taiwan	Mean: 32.8 Range: 25.0–46.0	10 (10 [100%] women)	Lower face. At 1 cm intervals on the masseter muscle	Masseter muscle	120 U per masseteric muscle, 20 U per 0.1 mL over six injections	One session	500 U per vial diluted in 2.5 mL of sterile distilled water to a concentration of 200 U/mL

Abbreviations: NR, not reported; RCT, randomized controlled trial; SD, standard deviation; U, units. ^1^ Reported for the whole cohort of 500 participants receiving treatment to several areas in the upper face in addition to lower face and neck. ^2^ Number of treatments to the platysma between June 2009 and February 2010.

**Table 2 toxins-13-00169-t002:** AbobotulinumtoxinA (aboBoNT-A) treatment approaches, by indication.

Regions of the Neck and Middle and Lower Areas of the Face	Doses Used in Individual Studies (Actual Dose or Mean/Range Dose Across Patients) ^1^	Number of Injection Points Used in Individual Studies ^2^
Marionette lines	18 U; 120‒250 U; 114.3 U; 154 U (Chang; Hexsel; Jabbour; Awaida)	2–4; 4; 150 (Jabbour; Chang; Awaida)
Jowls	114.3 U; 154 U (Jabbour; Awaida)	2–4; 150 (Jabbour; Awaida)
Nasolabial folds/nasal wrinkles	18 U; 120‒250 U (Chang; Hexsel)	4 (Chang)
Chin	18 U; 120‒250 U (Chang; Hexsel)	4 (Chang)
Perioral rhytids	18 U; 120‒250 U (Chang; Hexsel)	4 (Chang)
Neck volume/platysma	50–70 U; 114.3 U; 50–160 U; 154 U (Petchngaovilai; Jabbour; Hevia; Awaida)	2–4; 4–12; 150 (Jabbour; Hevia; Awaida)
Masseter muscle reduction	100–140 U; 120 U (Kim; Yu)	6 (Yu)
Malar wrinkles	120‒250 U (Hexsel)	NR (Hexsel)
Gummy/asymmetric smile	5–15 U; 120‒250 U (Mazzuco; Hexsel)	2–6 (Mazzuco)
Oral commissures	18 U; 114.3 U; 154 U (Chang; Jabbour; Awaida)	2–4; 4; 150 (Jabbour; Chang; Awaida)

Abbreviations: NR, not reported; U, units. References: Awaida et al. [26]; Chang et al. [27]; Hevia et al. [21]; Hexsel et al. [29,30]; Jabbour et al. [24]; Kim et al. [25]; Mazzuco and Hexsel [22]; Petchngaovilai et al. [23]; Yu et al. [23]. ^1^ Dose was reported in nine out of 10 included studies. ^2^ Number of injections was reported in seven out of 10 included studies.

**Table 3 toxins-13-00169-t003:** Summary of efficacy and safety outcomes.

	Efficacy			Safety	
Study	Assessment Methods	Clinical Effect	Key Findings	AEs Reported	Pain and/or Other Safety Findings
Awaida et al. 2018 [26]	Validated photonumeric scales. Investigator Global Aesthetic Improvement Scale used to assess improvement in the overall appearance of the lower face and neck	NR	There was statistically significant improvement in jowls (*p* < 0.0001), platysmal bands with contraction (*p* < 0.0001) and neck volume (*p* < 0.0008) 15 days post-treatment compared with pre-treatment There was no improvement in platysmal bands at rest, marionette lines and oral commissures	Injection-point ecchymosis lasting 2 days (n = 3 with micro injections of aboBoNT-A, n = 6 with Nefertiti technique) Mild dysphagia lasting 2 weeks (n = 1 with Nefertiti technique)	Mean (SD) VAS scores for pain from injections were 4.6 (2.3) for the micro injection technique and 0.6 (2.3) for Nefertiti technique (on scale of 0–10)
Chang et al. 2018 [27]	Digital image correlation	Improvement was observed at first 14-day follow-up Duration: 90 days (final follow-up)	At day 14, there were significant reductions in the magnitude of strain in the cheek (12%; *p* = 0.001), chin (7.8%; *p* = 0.022), marionette lines (17%; *p* < 0.001), upper lip (6.3%; *p* = 0.001) and perioral region as a whole (9.3%; *p* = 0.001). There was a 5.9% reduction in nasolabial folds (not statistically significant, *p* = 0.057) At day 14, there were significant increases in perioral volume in the nasolabial folds (*p* = 0.004), marionette lines (*p* = 0.006), upper lip (*p* = 0.004) and oral commissures (*p* < 0.001) There were further reductions in strain at day 90 There was no significant change in facial strain symmetry from baseline to day 90 By day 90, only the increase in volume in the marionette lines remained significant (*p* = 0.039), with volumes in the other three regions returning close to baseline levels	NR	No patients had any complications as a result of injections
Hevia 2010 [21]	NR	NR	NR	No AEs were reported for patients who received treatment	NR
Hexsel et al. 2013 [30]	Dermatological evaluation, wrinkle severity assessment, review of standardized photographs	Improvement was observed at first 4-week follow-up Duration: up to 20 weeks (participants reporting improvement in nasal wrinkles and lower eyelid wrinkles at follow-up)	There was a reduction in the severity of marionette lines between baseline and week 4 (*p* value not reported) At week 4, most of the participants presented at least 50% improvement in lower eyelid wrinkles, nasal wrinkles, perioral wrinkles and chin At week 16, more than 15% of the participants maintained at least 50% improvement in lower eyelid wrinkles, and more than 50% of the participants maintained at least 25% improvement in nasal and lower eyelid wrinkles At week 20, 18% of participants maintained at least 25% improvement in nasal wrinkles and 28% of the subjects maintained at least 25% improvement in lower eyelid wrinkles	Excessive perioral weakness (n = 30/77, AEs linked to injection dose) Lip asymmetry (n = 3) No serious AEs	Pain after injection was reported in two participants (although the area of face treated in these participants was not reported; thus, these may be participants that received treatment in the upper face)
Hexsel et al. 2013 [29]	NR	NR	NR	NR	NR
Jabbour et al. 2017 [24]	Validated photonumeric scales	Improvement was observed at 15- and 30-day follow-up visits Duration: NR	There was significant improvement in wrinkles/lines in the platysmal bands with contraction (*p* < 0.001) and rest (*p* < 0.009) No significant improvement observed in the jowls, marionette lines and oral commissures No significant improvement in neck volume scores	Injection-point ecchymosis (n = 5) Mild dysphagia and minor neck muscle weakness for 2 weeks post-injection (n = 1)	Mean (SD) VAS score for pain from injections was 1.2 (1.1) (on scale of 0–10)
Kim et al. 2005 [25]	Ultrasonogram	Onset: 2–4 weeks Duration: maximum effect was at 10–12 weeks	At 3 months, the mean thickness of the masseter muscle was reduced by 31%	Crunching power is decreased (n = 192) In crunching, muscle is protruded (n = 38) Unnatural smiling appearance (n = 8) Disappearance of facial dimple (n = 4)	NR
Mazzuco and Hexsel 2010 [22]	Clinician assessment of photographs (with the aid of two computer programs, the area of gum exposed was measured before and after treatment, to evaluate the level of improvement)	Onset: NR Duration: 3–5 months	A decrease in the degree of gum display was measured in all patients 20–30 days following treatment The average improvement of gingival exposure was: 75.09% in the overall sample 96% in those with anterior gummy smile 61.06% in those with posterior gummy smile 90.1% in those with mixed gummy smile 71.93% in those with asymmetric gummy smile	Asymmetric smile (n = 1) Difficulty in smiling (n = 1)	NR
Petchngao-vilai 2009 [23]	Assessment of photographs	Onset: In some cases instantaneous, but in most cases changes seen within 5–10 days Duration: 10–14 weeks	24.9% of participants (n = 65) attained high improvements with cheek lift, softening of nasolabial folds and re-defining of the facial contour ^1^ 65.52% of participants (n = 171) attained moderate improvements with cheek lift and facial contouring^1^ 9.58% of participants (n = 25) attained minimal improvements of facial contour ^1^	Minor facial asymmetry (n = 8)	NR
Yu et al. 2007 [28]	CT scan to measure muscle volume, patient-reported scores on a VAS to record facial change	Onset: 2 weeks Duration: NR	At 3 months, the volume of the masseter muscle was: decreased to 69.36% of baseline volume on the right side; decreased to 70.44% of baseline volume on the left side; and reduced by 30% overall (*p* < 0.001) There was no significant reduction in the volume of the other masticating muscles (temporalis, medial pterygoid, lateral pterygoid) compared with baseline (*p* > 0.001) Mean score reported by patients on facial improvement reached its maximum of 7.1 at 6 months One patient reported no change to facial appearance at any point during the study At the end of the study, only one patient reported a meaningful score of 8	Injection-point ecchymosis and swelling the day after injection that subsided 1 week later (n = 1) Soreness of bilateral masseters 1 day after the injection, which was aggravated when chewing food (n = 4) Easily fatigued while chewing food 2 days after the injection (n = 10) Bite weakness while eating vegetables or thick meat (n = 8) Less food intake because of more chewing effort required, but there was no interference to daily life (n = 1) Depression of the cheek on the right side (n = 2)	Mean VAS score 3 (on scale of 1–5) Nine participants reported a bearable discomfort during the injection

Abbreviations: AboBoNT-A, abobotulinumtoxinA; AE, adverse event; CT, computed tomography; NR, not reported; SD, standard deviation; VAS, visual analog scale. ^1^ Degrees of improvement were not defined.

## Data Availability

No new data were created or analyzed in this study. Data sharing is not applicable to this article.

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
