# Peer review of "Use of AbobotulinumtoxinA for Cosmetic Treatments in the Neck, and Middle and Lower Areas of the Face: A Systematic Review"

_toxins, 2021, doi:10.3390/toxins13020169_

Round 1
Reviewer 1 Report
Interesting and well-written review. The manuscript was adequately structured and the readability is very good. Although the final number of papers included in the revision is low and further research on the topic is required, I think the paper can be deemed worthy of publication.
Concerning the reviewed article: ABSTRACT Abstract was properly written, correctly summarizing the paper. Length is adequate. INTRODUCTION Introduction is adequate as well, providing clearly the background for the paper and summarizing in the last sentences the specific aim of the authors. RESULTS The Results section begins by schematically illustrating the way the literature review was performed. The study methodology is clear and logical; the readability of the section is thus ensured. The authors, in my opinion, report the most important data from each of the studies included in the review and at the same time correctly refrain from excessive descriptions which would eventually be detrimental for the readability of the paper. DISCUSSION The discussion section summarizes the main evidences emerging from the review and how they relate to other studies findings. A brief summary of the strengths and weaknesses of the paper is also provided. I do not have particular remarks on the tables and the images provided by the authors.Author Response
Please see the attachment

Reviewer 2 Report
A well-written review on use of BoNTA for Aesthetic treatment of lower face. Comments:
1) As the treatment location is very sensitive for severe adverse events such as dysphagia, I think the information in safety outcomes is essential and also deserves to be extended and perhaps include other references to other reports on adverse events in this treatment area.
Reviewer 3 Report
This systematic literature review is interesting, well-written and demonstrates at the first time describing the evidence on the treatment approaches, efficacy and safety outcomes associated with the use of aboBoNT-A specifically for cosmetic treatment of the neck, middle and lower areas of the face.
Unfortunately, this manuscript needs some important improvement and correction before publishing may be possible.
General points
Please correct in the whole manuscript all spaces between the words and all references numbers. For example, please say: Awaida et al. 2018 [15] instead of Awaida et al. 2018[15].
Please design as a separate Figure 3: AbobotulinumtoxinA dosis used for different indications. Please design this Figure 3 in the same way as Figure 2.
Please make as a separate Figure 4: The number of injection points of AbobotulinumtoxinA used for different indications. Please design this Figure 4 in the same way as Figure 2.
Special points
Abstract
Line 15: please say: reduction of masseter muscle.
Keywords
Please add also to keywords: middle face.
Introduction
Lines 29-30: please add more references at the end of this sentence.
Lines 30-32: please add more references at the end of this sentence.
Lines 32-33: please add references at the end of this sentence.
Lines 33-38: please add more references at the end of this sentence.
Lines 39-42: please add more references at the end of this sentence.
Lines 42-46: please add more references at the end of this sentence.
Lines 55-56: please add multiple references at the end of this sentence.
Results
Lines 69-70: you said: In the included studies, sample size ranged from 10 to 383 patients. Please also add to this sentence: the exact number of women (n=?) and men (n=?) and the middle age of all these patients.
Treatment approaches
Lines 92-100: Because of the exact injection points are very necessary for future injection outcomes and patients` satisfaction, please add also to this sentence information, which references were used and added in all these publications for topography of the different injection points.
Table 1
Please add also to your Table 1:
Please add to “Participants, n” column also the gender information, includes the exact number of women and men.
Please add also as a separate column: Results, Adverse effects and Assessment criteria described in all these studies.
Please say at the end of the table: Abbreviations: NR, not reported; RCT, randomized controlled trial; SD, standard deviation; U, units.
Efficacy outcomes
Lines 109-112: please add references at the end of this sentence.
Lines 132-136: please add also references for photonumeric scale.
Lines 139-141: please add references at the end of this sentence.
Lines 132-145: please add for all these studies additionally to patients´ numbers also gender information: the exact number of women and men.
Lines 141-142: you said: both demonstrated reductions in masseter muscle volume by approximately 30%. Please add which assessment criteria were used to access these results?
Safety outcomes
Line 149: please write out: AEs.
Lines 154-155: please add references for micro injections technique and for Nefertiti technique.
Patient satisfaction and quality of life
Lines 164-176: please add also which assessment criteria were used to access the satisfaction of the patients. It will be very fine also to compare the satisfaction between the woman and man.
Discussion
Lines 198-200: please add references at the end of this sentence.
Lines 201-202: please add references at the end of this sentence.
Lines 202-205: please add more references at the end of this sentence.
Lines 206-207: please add references at the end of this sentence.
Lines 195-207: please add for each these studies, how the patients´ satisfaction were measured.
Table S1, Table S2, Table S3
The same with Table 1, please add to each these tables: to “Participants, n” column also the number of women and men.
The same with Table 1, Please say at the end of the table: Abbreviations: …
